# Synergistic Effects of Ginsenoside Rb3 and Ferruginol in Ischemia-Induced Myocardial Infarction

**DOI:** 10.3390/ijms232415935

**Published:** 2022-12-14

**Authors:** Xu Chen, Tiantian Liu, Qiyan Wang, Hui Wang, Siming Xue, Qianqian Jiang, Junjun Li, Chun Li, Wei Wang, Yong Wang

**Affiliations:** 1College of Traditional Chinese Medicine, Beijing University of Chinese Medicine, Beijing 100029, China; 2School of Life Sciences, Beijing University of Chinese Medicine, Beijing 100029, China; 3School of Chinese Materia Medica, Beijing University of Chinese Medicine, Beijing 100029, China; 4Modern Research Center for Traditional Chinese Medicine, Beijing University of Chinese Medicine, Beijing 100029, China; 5Beijing Key Laboratory of TCM Syndrome and Formula, Beijing University of Chinese Medicine, Beijing 100029, China; 6School of Chinese Materia Medica, Guangzhou University of Chinese Medicine, Guangzhou 510006, China

**Keywords:** myocardial injury, ginsenoside Rb3, ferruginol, synergistic effect, RXRα

## Abstract

Previous research shows that ginsenoside Rb3 (G-Rb3) exhibit significant protective effects on cardiomyocytes and is considered a promising treatment for myocardial infraction (MI). However, how to improve its oral bioavailability and reduce its dosage remains to be studied. Previous studies suggest that Ferruginol (FGL) may have synergistic effects with G-Rb3. However, the underlying mechanisms remain to be explored. In this study, left anterior descending branch (LAD) coronary artery ligation or oxygen-glucose deprivation-reperfusion (OGD/R) were used to establish MI models in vivo and in vitro. Subsequently, the pharmacological effects and mechanisms of G-Rb3-FGL were explored by in vitro studies. The results showed that the G-Rb3-FGL co-treatment improved heart functions better than the G-Rb3 treatment alone in MI mice models. Meanwhile, the G-Rb3-FGL co-treatment can upregulate fatty acids oxidation (FAO) and suppress oxidative stress in the heart tissues of MI mice. In vitro studies demonstrated that the synergistic effect of G-Rb3-FGL on FAO, oxidation and inflammation was abolished by RXRα inhibitor HX531 in the H9C2 cell model. In summary, we revealed that G-Rb3 and FGL have a synergistic effect against MI. They protected cardiomyocytes by promoting FAO, inhibiting oxidative stress, and suppressing inflammation through the RXRα-Nrf2 signaling pathway.

## 1. Introduction

Myocardial infarction (MI) is one of the leading causes of death and disability worldwide [1,2]. The treatment goals for patients with MI are to improve their heart function, quality of life and to reduce mortality. Among many treatment methods, percutaneous coronary intervention (PCI) is the most effective way because it can protect the myocardium and reduce the infarct area. However, the process of myocardial reperfusion itself can paradoxically induce myocardial injury and cardiomyocyte death, which is referred to as “myocardial reperfusion injury” [3,4,5]. Therefore, there is an urgent need to develop novel or complement treatment strategies to improve the prognosis of patients with MI.

Retinoid X receptor α (RXRα) is a class of transcription regulatory factors, which is involved in the regulation of many biological processes, including cell proliferation, cell differentiation and pattern formation [6,7]. It plays an important role in cardiac development and cardiac muscle cell differentiation [8,9,10]. RXRα binds to DNA responsive elements in a ligand-dependent manner and is considered to be important among the nuclear receptor families [11,12]. Many reports have demonstrated that RXRα promotes energy metabolism and inhibits expressions of inflammatory enzymes, cytokines, chemokines, proteases and adhesion molecules by activating the transcription factor NF-E2 p45-related factor 2 (Nrf2) [12,13,14,15]. Nrf2 is also involved in suppression of oxidative stress by upregulating antioxidant enzymes [16]. Therefore, activation of the RXRα-Nrf2 pathway may protect against MI by promoting energy metabolism, suppressing oxidative stress and inhibiting inflammation.

Traditional Chinese medicine (TCM) has been used for thousands of years in the treatment of cardiovascular disease and is expected to be a complementary therapy for MI [17]. Ginsenosides, extracted from Panax notoginseng, are proved to be effective in treating cardiac hypertrophy, cardiac fibrosis, cardiomyocyte apoptosis and hypoxia/reoxygenation injury [18,19,20,21,22,23,24,25]. However, their applications are largely limited due to poor water solubility and side-effects at high doses [26]. Therefore, exploring ways to improve the oral bioavailability of ginsenosides and to lower the dosage by enhancing their effectiveness is of clinical significance.

According to the TCM theory, Panax notoginseng (Burkill) F.H.Chen (Sanqi in Chinese) and Salvia Miltiorrhiza Bunge (Danshen in Chinese), which are usually applied as a “herb-pair”, achieve a better efficacy in HF [27,28,29,30]. Ginsenoside Rb3 (G-Rb3) in Sanqi and Ferruginol (FGL) in Danshen are the effective components and have been proven to have significant cardio-protective effects. For example, our research found that G-Rb3 exerts a cardio-protective effect by inhibiting apoptosis and up-regulating energy metabolism [18]. FGL restores mitochondrial biogenesis and FAO for the treatment of doxorubicin-induced cardiotoxicity [31]. However, whether FGL can improve the effectiveness of G-Rb3 in the treatment of myocardial infarction remains unclear.

In this study, the efficacy of the G-Rb3-FGL co-treatment in improving cardiac function after MI were investigated in vivo. The pharmacological effects of G-Rb3-FGL on RXRα-Nrf2 pathway were explored in vivo and in vitro. This study aims to provide a new experimental basis for the clinical applications of G-Rb3 when combined with FGL to provide an insight into the treatment of MI.

## 2. Results

### 2.1. The Combination of G-Rb3 and FGL Improved Cardiac Function and Protected Cardiac Structure in LAD-Induced MI Mice

To assess the protective effects of the G-Rb3-FGL combination against MI injury in vivo, a LAD ligation-induced MI model was established in this study. As shown by echocardiography (Figure 1A,B), seven days after ligation, the ejection fraction (EF) and fraction shortening (FS) in the model group decreased when compared with the sham group (*p* < 0.001). Treatment with G-Rb3 can protect heart functions, as EF and FS were improved significantly (*p* < 0.01). Moreover, the EF and FS in the G-Rb3-FGL combination group were significantly higher than those in the G-Rb3 group, indicating that FGL could promote the protective effects of G-Rb3 (*p* < 0.05). H&E staining showed that in the model group cardiomyocytes at the border zone of infarction were arranged in a disordered way and inflammatory cell infiltration was observed (Figure 1C). Masson trichrome staining showed an abundant amount of collagen was deposited in the model group. The G-Rb3-FGL co-treatment can ameliorate collagen deposition and improve cardiomyocyte structures (Figure 1D).

### 2.2. The Combination of G-Rb3 and FGL Alleviated Dyslipidemia and Oxidative Stress in LAD-Induced MI Mice

Dyslipidemia was developed in LAD-induced MI mice, as illustrated by the increased serum TG, T-CHO, LDL-C levels, and a reduced HDL-C level. Treatment with G-Rb3 significantly improved dyslipidemia (*p* < 0.05). Moreover, the effects of the G-Rb3-FGL co-treatment were more significant than those of the G-Rb3 treatment alone (*p* < 0.05, Figure 2A). In addition, the plasma MDA level was higher and SOD activity was lower in mice of the model group when compared with those of the sham group (*p* < 0.001). When compared with the G-Rb3 group, G-Rb3-FGL further reduced the MDA level (*p* < 0.05) and increased the SOD level (*p* < 0.01) (Figure 2B,C). A Western blot analysis showed that G-Rb3-FGL promoted the expressions of RXRα and Nrf2 and inhibited the expression of p65-NF-κB (Figure 2D), indicating that G-Rb3-FGL may play a cardioprotective role through the RXRα-Nrf2 pathway.

### 2.3. G-Rb3-FGL Protected H9C2 Cells against OGD/R-Induced Cellular Injury

The effective doses of each compound were explored by CCK8 tests. As shown in Figure 3A–F, treatment with G-Rb3 (20~80 μM) and FGL (1~2 μM) protected against OGD/R-induced H9C2 cell injury. Impressively, the G-Rb3 (10 μM)-FGL (1 μM) combination had better protective effects than that of the G-Rb3 (40 μM) treatment (Figure 3G). In the subsequent experiment, we chose G-Rb3 (10 μM)-FGL (1 μM) as a treatment concentration to explore the pharmacological mechanisms.

### 2.4. Effects of G-Rb3-FGL Co-Treatment on Fatty Acid Oxidization in OGD/R-Stimulated H9C2 Cells

TG is often used as an indicator of fatty acid metabolism. We tested the TG level to evaluate the fatty acid metabolism in vitro. Treatment with G-Rb3-FGL suppressed the increase in TG levels induced by OGD/R (*p* < 0.05). Such change was significantly (*p* < 0.01) suppressed by the co-incubation with HX531, the inhibitor of RXRα (Figure 4A). Since RXRα can regulate the energy metabolism of cardiomyocytes, ATP production and mitochondrial membrane potentials were then measured. As shown in Figure 4B, ATP levels were significantly up-regulated in the G-Rb3-FGL co-treatment group when compared with the model group (*p* < 0.001). JC-1 staining showed that treatment with G-Rb3-FGL can increase the mitochondrial membrane potential (a lower level of green and higher level of red fluorescence intensity, *p* < 0.01). Intriguingly, co-treatment of cells with HX531 attenuated the protective effects of G-Rb3-FGL on mitochondrial membrane potentials (Figure 4C). These results suggested that G-Rb3-FGL co-treatment promoted the energy metabolism in H9C2 cells through activating RXRα. Furthermore, the downstream molecules of RXRα, which regulate lipid metabolism, were detected by a Western blot analysis. The results showed that the expressions of CD36, CPT-1A and ACADL were activated by the G-Rb3-FGL co-treatment. Moreover, the effects of the G-Rb3-FGL co-treatment on these proteins were attenuated by RXRα inhibitor HX531, suggesting that the regulative effects of G-Rb3-FGL on lipid metabolism was mediated through RXRα (Figure 4D).

### 2.5. Effects of G-Rb3-FGL Co-Treatment on Oxidative Stress and Inflammation Response in OGD/R-Stimulated H9C2 Cells

Mitochondrial energy metabolic disorder induced by OGD/R can lead to oxidative stress, which will further cause an inflammatory response. As shown in Figure 5A,B, the results of DCFH-DA staining showed that ROS levels were significantly higher in the OGD/R group than those in the control group. After treatment with G-Rb3-FGL, intracellular ROS was significantly reduced when compared with those in the OGD/R group. However, the anti-oxidant effect of G-Rb3-FGL was attenuated by the RXRα inhibitor HX531. As a key nuclear transcription factor, RXRα promotes the migration of Nrf2 to the nucleus and the expression of the antioxidant enzyme HO-1, which is involved in antioxidant and inflammatory reaction responses [14]. As shown in Figure 5C, OGD/R stimulation promoted p65-NF-κB expression and nuclear translocation when compared with untreated cells, whereas the G-Rb3-FGL treatment reversed this change. In addition, WB results showed that G-Rb3-FGL can effectively promote Nrf2 expression, inhibit p65-NF-κB activation and suppress the expressions of COX2, IL-1β and TNF-α. Interestingly, the regulative effect of G-Rb3-FGL was eliminated after administration with the RXRα inhibitor HX531 (Figure 5E,F). These in vitro results illustrated that the anti-oxidant and anti-inflammatory effect of G-Rb3-FGL are related to the RXRα-mediated Nrf2-NF-κB pathway.

### 2.6. G-Rb3-FGL Inhibited Apoptosis via RXRα In Vitro

Prolonged oxidative stress and inflammation will eventually lead to cell death. Hoechst staining indicated that OGD/R stimulation induced H9C2 cell apoptosis and the G-Rb3-FGL co-treatment can suppress cellular apoptosis, whereas HX531 can abolish the protective effect of G-Rb3-FGL (Figure 6A). The decreased expression of Bcl-2 and the increased expression of Bax in OGD/R injury-induced cells were observed (Figure 6B,C). The G-Rb3-FGL co-treatment increased the expression of Bcl-2 and reduced the expression of Bax. The effects of G-Rb3-FGL on these proteins were attenuated by RXRα inhibitor HX531.

### 2.7. FGL Had the Potential to Improve the Oral Bioavailability of G-Rb3

P-glycoprotein (P-gp), also known as multidrug resistance protein 1 (MDR1), is an efflux transporter that affects the absorption, distribution and elimination of a variety of compounds. Rh123, a known P-gp substrate, was added to the Caco-2 cells to evaluate the impact of any potential interactions of G-Rb3 and FGL on pharmacokinetics. Verapamil is one of the widely recognized P-gp inhibitors and it was applied as a control in this study [32]. We found that 40~400 μM G-Rb3 and 1~50 μM FGL were non-toxic to Caco-2 cells (Figure 7A). The G-Rb3-FGL co-treatment can significantly down-regulate the expression of P-gp protein when compared with the G-Rb3 treatment alone (Figure 7B). In the meantime, both FGL and Verapamil increased the accumulation of Rh123 in Caco-2 cells (Figure 7C).

## 3. Discussion

In this study, we found that G-Rb3-FGL has a synergistic protective effect against MI. G-Rb3 combined with FGL treatment can improve the cardiac function in MI mice. G-Rb3-FGL also enhanced cell viability in OGD/R-stimulated H9C2 cells. In addition, we found that FGL has the potential to improve the oral bioavailability of G-Rb3 effectively and reduce the dosage of G-Rb3. G-Rb3-FGL exerts protective effects by promoting lipid metabolism, increasing antioxidant capacity and decreasing the inflammatory response through the RXRα-Nrf2 pathway (Figure 8).

MI results from the irreversible death of cardiomyocytes which caused by prolonged hypoxia or inadequate blood supply [33]. Due to acute ischemia and hypoxia, myocardial cells induce the production of reactive oxygen species (ROS). Oxidative stress stimulates the production of inflammatory factors, which in turn promote the formation of ROS, leading to the cascade amplification of oxidative stress and inflammatory response. Therefore, anti-oxidation and anti-inflammation are considered to be important means of managing MI. As an important nuclear transcription factor, RXRα plays a key role in promoting lipid metabolism and suppressing inflammation [12,13,34]. In addition, RXRα can indirectly regulate NF-κB through Nrf-2 [35]. In this study, we explored whether traditional Chinese medicine could exert myocardial protective effects by regulating the RXRα-Nrf2 signaling pathway.

Our previous studies showed that the combination of Danshen and Sanqi can significantly improve the cardiac function of rats after MI by regulating energy metabolism and promoting angiogenesis [27,36]. G-Rb3, the main component of Sanqi, was shown to exert a protective effect on the myocardium by improving energy metabolism, suppressing oxidative stress and improving microcirculation disorders [18,20,37]. FGL is a natural polyphenol and terpenoid compound that was shown to play a cardioprotective role in models of doxorubicin-induced myocardial injury [31]. Owing to the complexity of serious diseases, such as myocardial infarction, recent developments have gradually shifted from a focus on monotherapy to combined or multiple therapies because the synergy of therapeutic agents or techniques results in ostentatious super additive (namely “1 + 1 > 2”) therapeutic effects [38]. However, it is not clear whether the two compounds have a synergistic effect in MI. In this study, we found that the G-Rb3-FGL co-treatment can promote FAO, reduce oxidative stress and suppress the inflammation response partly through the RXRα-Nrf2 pathway.

Reduced levels of FAO is generally considered to be the most critical feature of changes in energy metabolism during MI [39]. Therefore, improving mitochondrial FAO is a potential therapeutic strategy of MI. Our previous study found that G-Rb3 had cardioprotective effects by regulating energy metabolism through the PPARα/RXRα pathway [18]. In this study, we confirmed that the G-Rb3-FGL had better effects in improving cardiac function and dyslipidemia than G-Rb3 treatment alone. In vitro, JC-1 staining suggested that G-Rb3-FGL can effectively improve the mitochondrial membrane potential levels and this change was mediated by RXRα. In addition, we found that the G-Rb3-FGL co-treatment up-regulated the expressions of key enzymes during FAO, including CD36, CPT-1A and ACADL, and this effect was attenuated by the RXRα inhibitor (Figure 4). Taken together, these results suggest that the effect of G-Rb3-FGL in promoting FAO is mediated by RXRα.

In addition to energy metabolic disorders, oxidative stress and inflammatory responses also aggravate the progression of MI. As an important nuclear transcription factor, RXRα plays an important role in regulating various nuclear hormone signaling pathways. It has been confirmed that RXRα can inhibit the inflammatory response by inhibiting the release of proinflammatory factors, so as to protect against hypotension and tachycardia caused by sepsis [40]. Moreover, RXRα signaling protects cardiomyocytes from hyperglycemia by reducing oxidative stress [41]. Nrf2 plays a crucial role in combating various oxidative stress and heart remodeling after MI [42]. In addition, Nrf2 upregulates expression of HO-1 and downregulate NF-κB, which in turn suppressed pro-inflammatory factors, including IL-1β and TNF-α [43,44]. Recently, it has been confirmed that RXRα can directly target and regulate Nrf2 in cancer. However, whether RXRα can directly regulate Nrf2 in cardiovascular diseases has not been reported [14]. In this study, we evaluated whether the G-Rb3-FGL co-treatment could suppress oxidant stress and inflammatory responses through the RXRα-Nrf2 pathway. It turned out that the G-Rb3-FGL co-treatment can regulate the expression of RXRα and Nrf2 in the OGD/R-stimulated H9C2 cell model. After the cells were co-treated with the RXRα inhibitor, the anti-oxidative and anti-inflammatory effects of G-Rb3-FGL were attenuated. Moreover, the anti-apoptotic effect of G-Rb3-FGL was also reduced. These results suggest that the synergistic effect of G-Rb3-FGL may be mediated by the RXRα-Nrf2 signaling pathway. To our knowledge, this study illustrated for the first time that activating RXRα may serve as a novel therapeutic strategy for MI.

Furthermore, we investigated whether FGL has the potential to increase the oral bioavailability of G-Rb3. Intestinal P-gp is located in the apical membrane of its epithelial cells and is recognized as a major factor affecting the kinetics of drugs that are taken orally. The increase in P-gp function is associated with the decrease in drug efficiency (drug resistance) [45]. Inhibition of P-gp can effectively prevent the drug pump effect of P-gp, reduce drug efflux and improve drug bioavailability in vivo. In the current study, we explored the oral bioavailability of G-Rb3-FGL by detecting P-gp protein levels and Rh123 (a widely used index of P-gp-mediated transporter) in Caco-2 cell lines. The results demonstrated that the G-Rb3-FGL co-treatment can maintain P-gp at a lower level when compared with the G-Rb3 treatment alone (Figure 7). In addition, our results also proved that FGL can promote Rh123 accumulation. The combined use of drugs that inhibit P-gp may increase the systemic exposure of P-gp substrates, thus affecting the pharmacokinetics of various compounds [46]. Collectively, our data suggested that FGL has the potential to increase the oral bioavailability of G-Rb3 by inhibiting P-gp. The detection of oral bioavailability is an essential step for the discovery of new drugs to successfully enter the pre-clinical development stages. In the future, we will choose appropriate methods to conduct in-depth discussion on the oral availability of G-Rb3-FGL.

## 4. Materials and Methods

### 4.1. MI Model Establishment and Animal Grouping

All Adult male ICR mice experiments conformed to the Guide for the Institutional Animal Care and Use Committee (IACUC) and were approved by the Animal Care Committee of Beijing University of Chinese Medicine. To induce myocardial infarction in mice, the left anterior descending (LAD) coronary artery was subjected to ligation according to the previous studies [47,48]. After the establishment of the MI model, all the mice were divided into five groups randomly: (1) sham group (*N* = 9); (2) model group (*N* = 9); (3) G-Rb3 group (*N* = 9, 0.63 mg/kg/d); (4) G-Rb3-FGL co-treatment group (*N* = 9, 0.32 mg (G-Rb3)/kg/d + 0.032 mg (FGL)/kg/d); (5) fenofibrate positive control group (*N* = 9, 50 mg/kg/d). G-Rb3 was purchased from Shanghai yuanye Bio-Technology, Shanghai, China. FGL was purchased from Shanghai yuanye Bio-Technology, Shanghai, China. In the G-Rb3-FGL co-treatment group, the ratio of G-Rb3 to FGL was 10 to 1. Mice in the sham operation group and the model group were treated with the same volume of carboxymethyl cellulose sodium. Animals in all the above groups were treated by gavage for seven days.

### 4.2. Cardiac Functions and Histological Assessment

The heart functions of mice were examined by echocardiography through a high-resolution imaging system (Vevo 3100 imaging system, Visual Sonics, Bothell, WA, USA). The detailed procedure was described previously [49].

Cardiac tissues were collected for histological analysis. Left ventricle tissues from each group were fixed with 4% paraformaldehyde, embedded with paraffin, and sectioned at 5 μm slices. Sections were stained with hematoxylin–eosin (H&E) and Masson’s stain, and were then examined under an optical microscope (Leica Microsystems GmbH, Leica Microsystems Inc., Deerfield, IL, USA).

### 4.3. Serum Detection

Serum lipids, such as total cholesterol (TG), triglycerides (T-CHO), high-density lipoprotein (HDL-C), low-density lipoprotein (LDL-C), superoxide dismutase (SOD) and malondialdehyde (MDA) were detected by an enzyme-linked immunosorbent assay.

### 4.4. Cell Culture

The oxygen-glucose deprivation-reperfusion (OGD/R) stimulated cardiomyocyte injury model was established according to previous studies [18,50]. In this study, H9C2 cells were divided into four groups: control group, OGD/R-induced model group, G-Rb3 (10 μM)-FGL (1 μM) treatment group and G-Rb3-FGL+HX531 (1 μM) treatment group. HX531 is the inhibitor of RXRα. After growing up to 80% confluence, the cells were cultured at Earle’s balanced salt solution with hypoxic settings for 8 h, followed by reoxygenation for 12 h. H9C2 cells in G-Rb3-FGL group were treated in the same way, except that the Earle’s balanced salt solution and culture media contained G-Rb3 (10 μM) and FGL (1 μM). Moreover, HX531 was co-cultured with G-Rb3 in the G-Rb3-FGL+HX531 group. In the control group, fresh nutrient solution was applied to treat cells at the time points of 8 and 12 h.

The human colon cancer, Caco-2 cells were maintained in Minimum Essential Medium (Gibco, Waltham, MA, USA) containing 20% fetal bovine serum (FBS, Gibco, Waltham, MA, USA) and 1% penicillin/streptomycin (purchased from Gibco, Waltham, MA, USA) at 37 °C in a humidified (37 °C and 5% CO_2_) incubator. Caco-2 cells were divided into four groups: control group, G-Rb3 (160 μM) treatment group, G-Rb3 (160 μM)-FGL (10 μM) treatment group and G-Rb3 (160 μM)-Verapamil (10 μM) (Sigma, St. Louis, MO, USA) treatment group. After growing up to 80% confluence, the cells were cultured with a cell medium containing G-Rb3, G-Rb3-FGL and G-Rb3-Verapamil (10 μM) (Sigma, USA), respectively, for 24 h.

### 4.5. Intracellular ROS Detection

H9C2 cells were seeded at a density of 5 × 10^4^ cells/mL in 12-well plates. After OGD/R and G-Rb3-FGL or G-Rb3-FGL+HX531 treatments, H9C2 cells were washed with PBS three times. The intracellular ROS level was measured by a 2′,7′-dichlorodihydrofluorescein diacetate (DCFH-DA) assay kit (S0033S, Beyotime Biotechnology, Shanghai, China) following the manufacturer’s instructions. In short, H9C2 cells were incubated with 50 μM DCFH-DA at 37 °C in the dark for 30 min. Images were obtained by a fluorescence microscope (Olympus IX51, Tokyo, Japan). Image J software was used (National Institutes of Health) to analyze the average fluorescence intensity.

### 4.6. ATP and Mitochondrial Membrane Potential Assessment

ATP was measured using an ATP assay kit (S0026, Beyotime Biotechnology, Shanghai, China) according to the manufacturer’s instructions. The luminescence produced was measured with a luminometer counter (perkin-elmer, Waltham, MA, USA), and the concentration of the ATP was calculated using an ATP standard curve. The mitochondrial membrane potential was detected and quantified in live cells by a laser scanning confocal microscope with the JC-1 assay Kit (Abnova, KA1324, CA, USA). The relative ratio of red/green fluorescence was used as an index of membrane potential. A reduced ratio of red/green fluorescence indicates that the membrane potential is decreased.

### 4.7. Detection of Intracellular Rh123 Content

To observe the influence of modulators on Rh123 accumulation in Caco-2 cells, 5 × 10^4^/mL cells were loaded on to laser confocal dishes and cultured overnight in the incubator. The medium was removed and cells were washed with PBS, a new medium containing 5 μM Rh123 was then added and incubated for 30 min, the cells were then washed twice with PBS before examining with the confocal laser scanning microscope (Olympus IX51, Tokyo, Japan). Data were collected and analyzed by the Image J software (1.43 Version) (National Institutes of Health).

### 4.8. Western Blot Analysis

Protein samples were obtained from the lysates of cultured cells and the protein concentration was determined using the BCA protein detection system. Proteins were then denatured in a 4 × loading buffer at 100 °C for 10 min and separated by 12.5% SDS-PAGE. Proteins were transferred from the gels onto PVDF membranes. The antibodies were as follows:
**Antibodies****Source****Identifier**anti-RXRα antibodyAbcam, Cambrige, UKCat#ab125001anti-CD36 antibodyAbcam, Cambrige, UKCat#ab252922anti-CPT1A antibodyAbcam, Cambrige, UKCat#ab128568anti-ACADL antibodyAbcam, Cambrige, UKCat#ab196655anti-p65-NF-κB antibodyAbways, Shanghai, ChinaCat#CY5034anti-TNF-α antibodyAbcam, Cambrige, UKCat#ab307164anti-IL-1β antibodyProteinTech Group, Rosemont, IL, USACat#10663anti-COX2 antibodyAbcam, Cambrige, UKCat#ab179800anti-Bax antibodyAbcam, Cambrige, UKCat#ab32503anti-Bcl-2 antibodyAbcam, Cambrige, UKCat#ab196495anti-HO-1 antibodyAbways, Shanghai, ChinaCat#CY5113anti-Nrf-2 antibodyAbways, Shanghai, ChinaCat#CY5136

### 4.9. Statistical Analysis

The results were analyzed by GraphPad Prism 8.0 and data were expressed as mean ± SD. One-way analysis of variance analysis (ANOVA) and Dunnett’s test were used to compare the differences between two or more groups. The difference was considered a statistical significance when *p* < 0.05.

## 5. Conclusions

This study shows that G-Rb3-FGL combination therapy can improve the in vitro and in vivo effects of MI. The synergistic effect of G-Rb3 and FGL on MI is mediated by promoting FAO, inhibiting oxidative stress and inhibiting the inflammatory response through the RXRα-Nrf2 signaling pathway. In addition, FGL increased the oral bioavailability of G-Rb3. These findings provide new insights into the molecular mechanism of the protective effect of G-Rb3-FGL therapy in MI, and may contribute to future research on the possible interaction of therapeutic drugs in the treatment of MI.

## Figures and Tables

**Figure 1 ijms-23-15935-f001:**
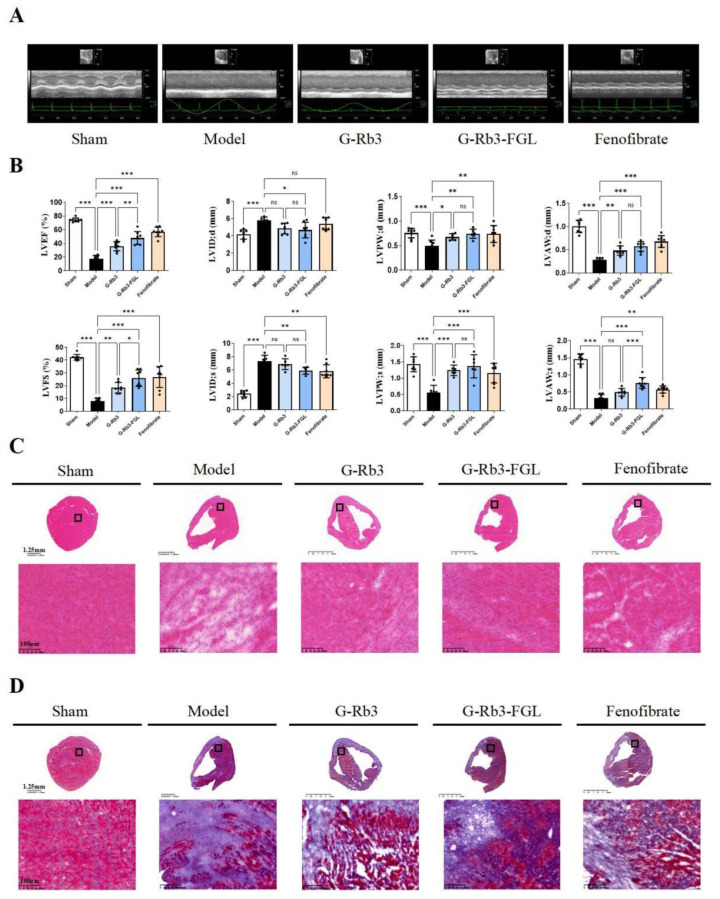
G-Rb3-FGL co-treatment improved heart function and ameliorated pathological changes in MI mice. (**A**,**B**) Echocardiographic analysis. The levels of LVEF, LVFS, LVID; d, LVID; s with G-Rb3, G-Rb3-FGL treatment were recorded by echocardiography. (**C**,**D**) H&E and Masson staining. ^ns^ No statistical significance, * *p* < 0.05, ** *p* < 0.01, *** *p* < 0.001 vs. model group. *N* = 8 per group.

**Figure 2 ijms-23-15935-f002:**
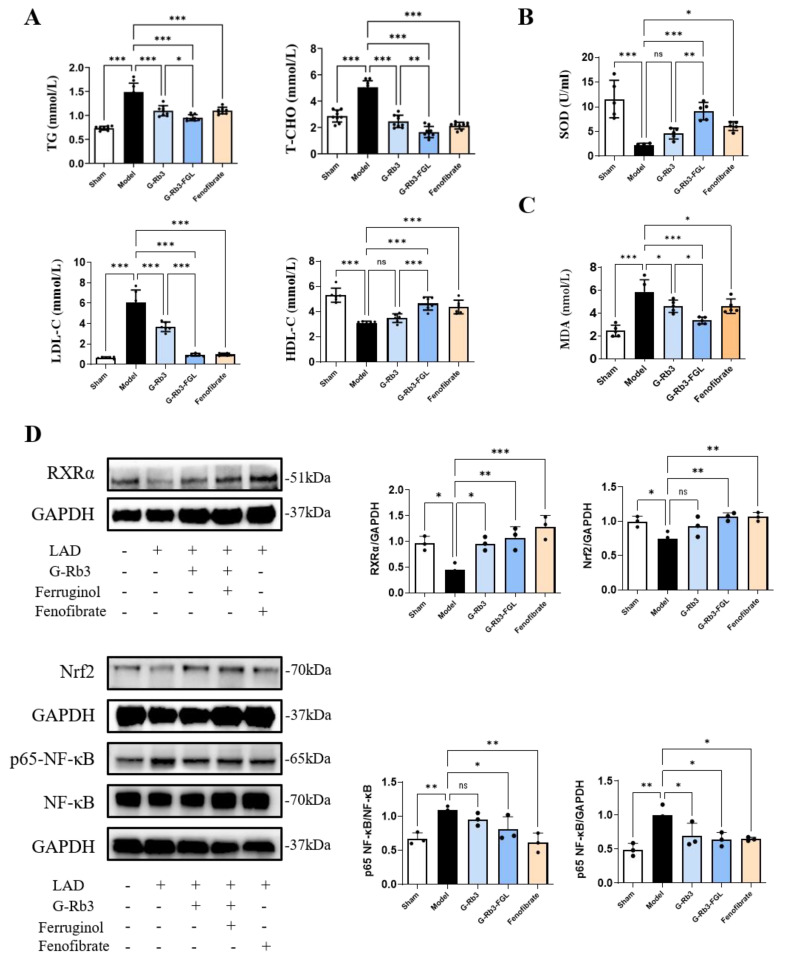
G-Rb3-FGL co-treatment alleviated dyslipidemia and oxidative stress in MI mice. (**A**) Levels of TG, T-CHO (*N* = 9 per group) and HDL-C, LDL-C (*N* = 6 per group) in serum. (**B**,**C**) Levels of SOD and MDA in serum. *N* = 5 per group. (**D**) The relative protein levels of RXRα, Nrf2 and p65-NF-κB in heart. All data were presented as means ± SD from independent experiments performed in triplicate. ^ns^ No statistical significance, * *p* < 0.05, ** *p* < 0.01, *** *p* < 0.001 vs. model group. *N* = 3 per group.

**Figure 3 ijms-23-15935-f003:**
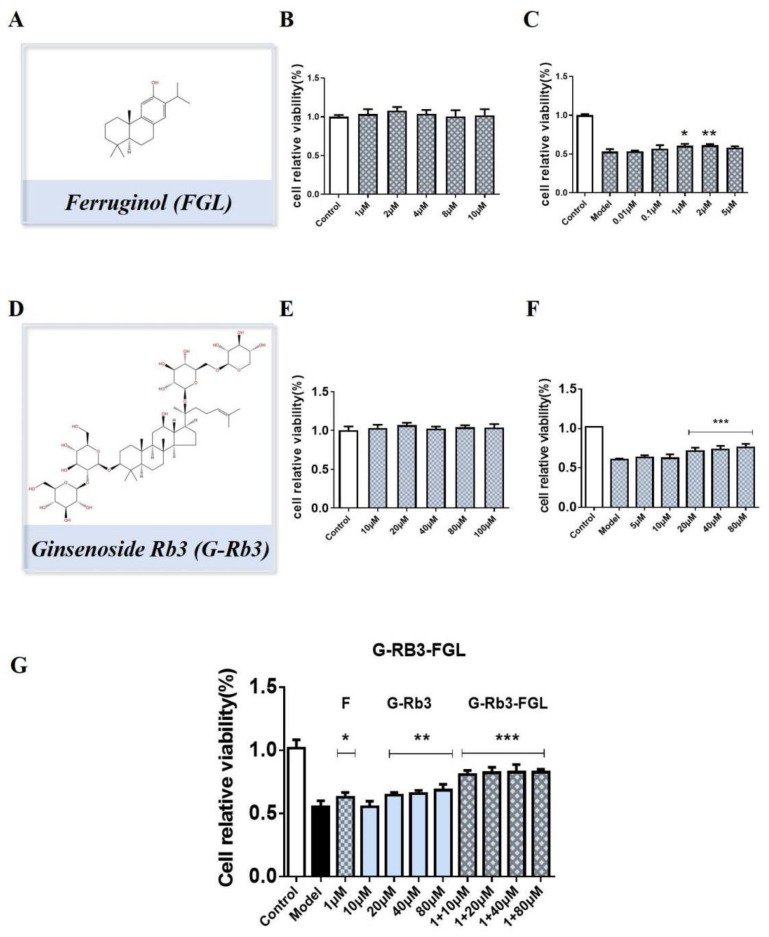
G-Rb3-FGL have synergistic effect. (**A**–**F**) Cell relative viability of G-Rb3 and FGL alone in OGD/R injured H9C2 cells. (**G**) Cell relative viability of G-Rb3 and FGL combination in OGD/R injured H9C2 cells. ^ns^ No statistical significance, * *p* < 0.05, ** *p* < 0.01, *** *p* < 0.001 vs. model group. *N* = 6 per group.

**Figure 4 ijms-23-15935-f004:**
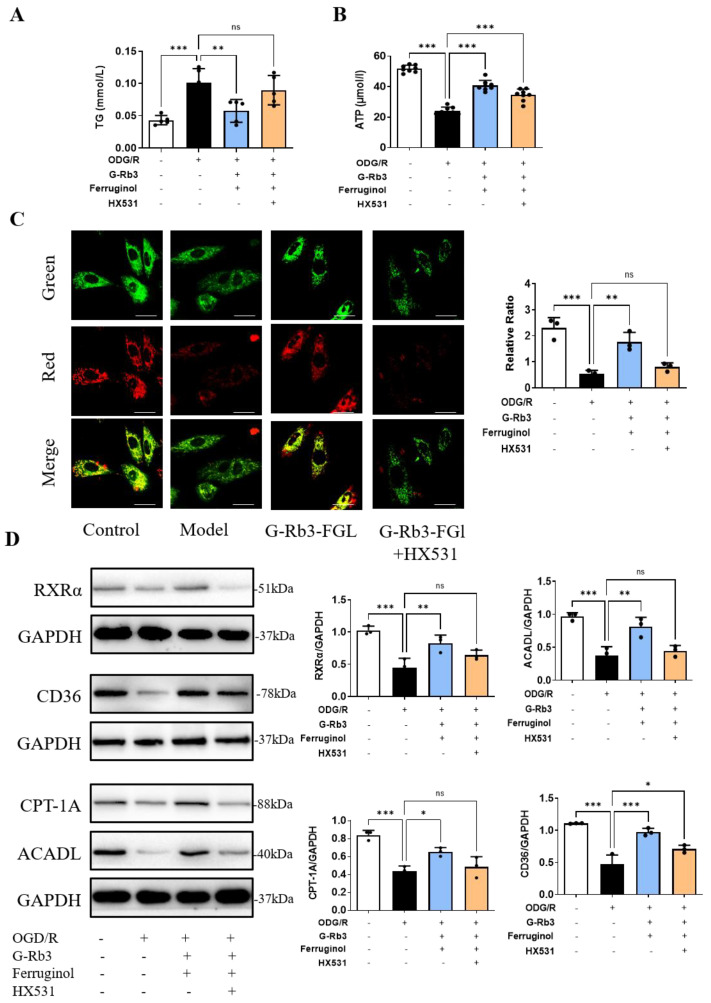
G-Rb3-FGL promoted fatty acid oxidization in vitro. (**A**,**B**) G-Rb3-FGL treatment changed the level of TG (*N* = 5 per group) and ATP (*N* = 8 per group) in H9C2 cells subjected to OGD/R. (**C**) JC-1 staining of four groups (×100), red fluorescence of the J-aggregates and green fluorescence of the J-monomeric. (**D**) The expression level of fatty acid oxidization-associated proteins in H9C2 cells quantified and shown as relative protein expression levels after normalization to GAPDH (*N* = 3 per group). All data were presented as means ± SD from independent experiments performed in triplicate. ^ns^ No statistical significance, * *p* < 0.05, ** *p* < 0.01, *** *p* < 0.001 vs. model group.

**Figure 5 ijms-23-15935-f005:**
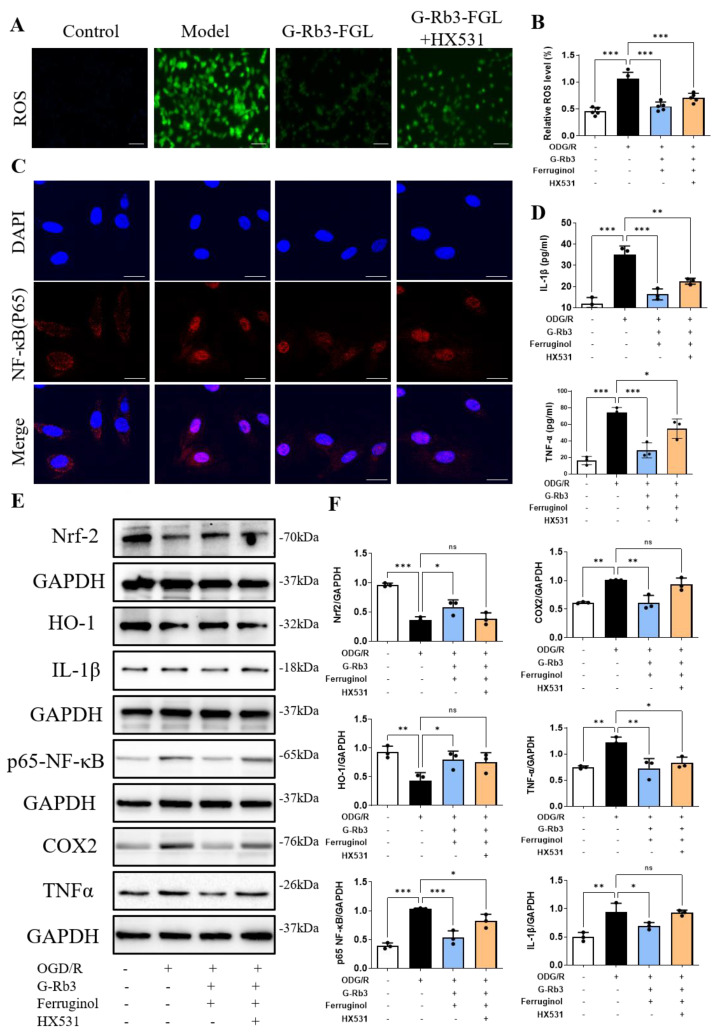
G-Rb3-FGL co-treatment exerted anti-oxidant and anti-inflammatory effects in OGD/R stimulated cell. (**A**,**B**) ROS (×10). Green fluorescence of the ROS form of the dye. (**C**) G-Rb3-FGL suppressed the nuclear localization of p65-NF-κB in OGD/R-stimulated H9C2 cells (×100). Red fluorescence of the NF-κB (p65) and blue fluorescence of the nucleus. (**D**) The release of IL-1β and TNF-α in H9C2 cell supernatants. (**E**,**F**) Representative immunoblots and the quantification of Nrf2, HO-1, p65-NF-κB, COX2, IL-1β and TNF-α in different groups. *N* = 3 per group. ^ns^ No statistical significance, * *p* < 0.05, ** *p* < 0.01, *** *p* < 0.001 vs. model group.

**Figure 6 ijms-23-15935-f006:**
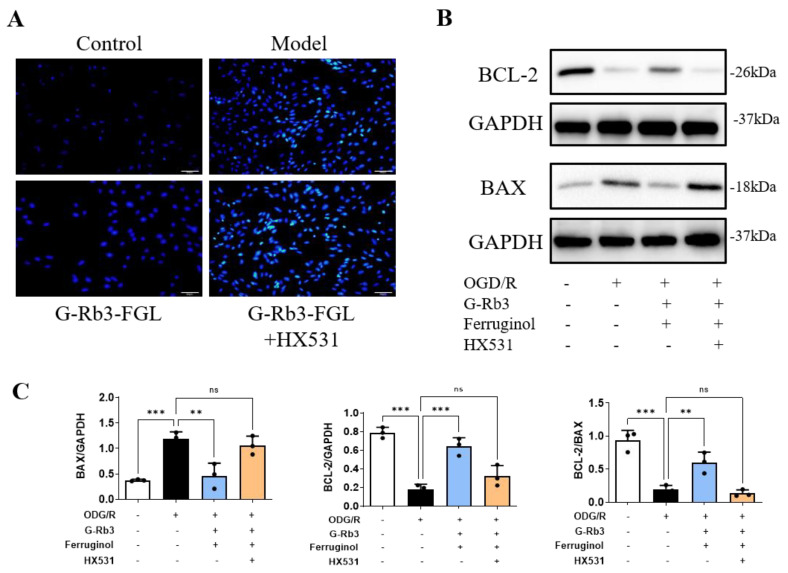
G-Rb3-FGL inhibits apoptosis via RXRα in vitro. (**A**) The results of Hoechst staining (×10). Blue fluorescence represents apoptotic cells. (**B**,**C**) Representative immunoblots and the quantification of Bax and Bcl-2 in different groups. *N* = 3 per group. ^ns^ No statistical significance, ** *p* < 0.01, *** *p* < 0.001 vs. model group.

**Figure 7 ijms-23-15935-f007:**
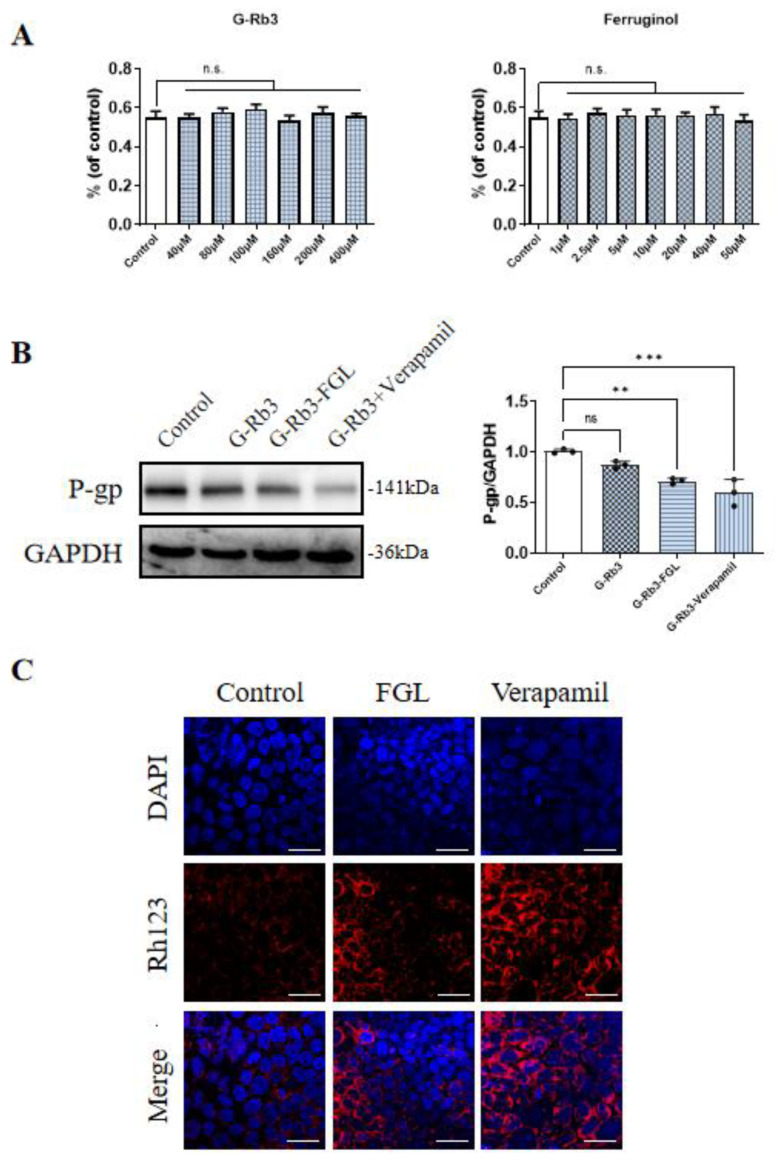
G-Rb3-FGL co-treatment inhibited P-gp expression. (**A**) Cell relative viability of G-Rb3 and FGL alone in Caco-2 cells. (**B**) Representative immunoblots and the quantification of P-gp in different groups. *N* = 3 per group. (**C**) The results of Rh123 staining (×100). Red fluorescence of the Rh123 and blue fluorescence of the nucleus. ^ns^ No statistical significance, ** *p* < 0.01, *** *p* < 0.001 vs. model group.

**Figure 8 ijms-23-15935-f008:**
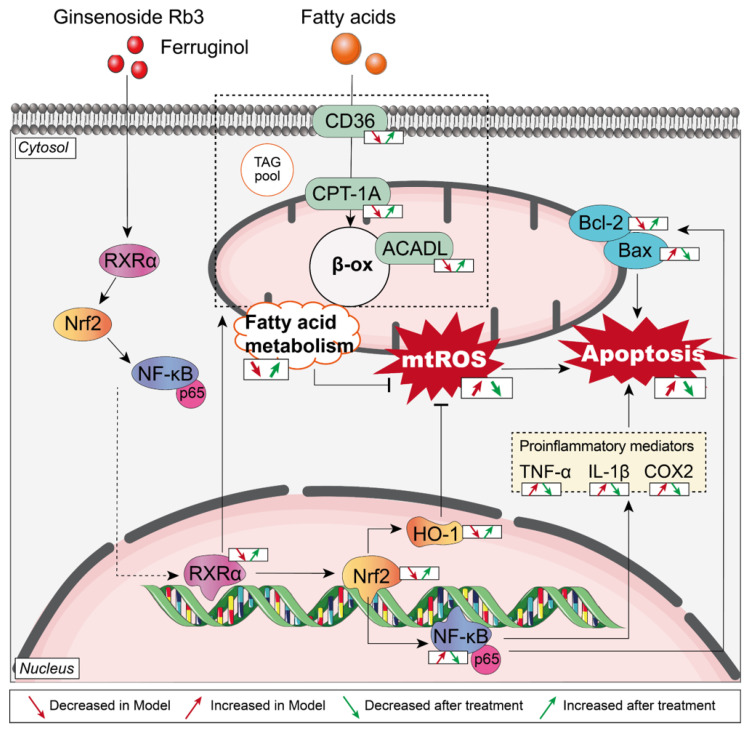
Mechanism of G-Rb3-FGL combination in the treatment of MI.

## Data Availability

Not applicable.

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
