# Peer review of "Synergistic Effects of Ginsenoside Rb3 and Ferruginol in Ischemia-Induced Myocardial Infarction"

_ijms, 2022, doi:10.3390/ijms232415935_

Round 1
Reviewer 1 Report
the experimental study was organized with methodological rigor and was of such interest as to foresee the reproducibility of the data for expanding the results. objectives and goals, step by step, are satisfied by the results.
Author Response
Point 1: the experimental study was organized with methodological rigor and was of such interest as to foresee the reproducibility of the data for expanding the results. objectives and goals, step by step, are satisfied by the results.
Response 1: Thank you for your recognition of the content of this study.

Reviewer 2 Report
The manuscript "Synergistic effects of Ginsenoside Rb3 and Ferruginol in ischemia-induced myocardial infarction" by Chen et al. is very detail and important study. The manuscript is well written. I suggest a few minor corrections:
Abstract, first sentence start with Previous research show that ginsenoside...., or similar...
Abstract, line 19 - MI? Add myocardial infarction (MI)
Figure 3. - equalize the size of the benzene rings in ferruginol and Ginsenoside Rb3
Conclusion - start with This study show that G-Rb3-FGL combination therapy... or similar
Author Response
The manuscript "Synergistic effects of Ginsenoside Rb3 and Ferruginol in ischemia-induced myocardial infarction" by Chen et al. is very detail and important study. The manuscript is well written. I suggest a few minor corrections:
Point 1: Abstract, first sentence start with Previous research show that ginsenoside...., or similar....
Response 1: Thank you for your comment. We have modified the corresponding sentences in the article according to your comment.
Point 2: Abstract, line 19 - MI? Add myocardial infarction (MI)
Response 2: Thank you for your careful review of our manuscript. We have modified it in the article.
Point 3: Figure 3. - equalize the size of the benzene rings in ferruginol and Ginsenoside Rb3
Response 3: Thank you for your valuable comment. We have modified the figure according to your opinion. We equalized the size of the benzene rings in FGL and G-Rb3, and marked the modified places.
Point 4: Conclusion - start with This study show that G-Rb3-FGL combination therapy... or similar
Response 4: Thank you for your comment. We apologize for our poor language of manuscript. We have modified the corresponding sentences in the article.

Reviewer 3 Report
Review
This work examines the efficacy of G-Rb3-FGL co-treatment in improving cardiac function 72 after MI were investigated in vivo. The pharmacological effects of G-Rb3-FGL on RXRα- 73 Nrf2 pathway were explored in vivo and in vitro. Through left anterior descending branch (LAD) coronary artery ligation or oxygen- 22 glucose deprivation-reperfusion (OGD/R) were used to establish MI models in vivo and in vitro. To verify the pharmacological effects and mechanisms of G-Rb3-FGL were explored by in vitro 24 studies.
This research is both practical and novel. But, some of the data in the paper are confusing, so several modifications and explanations are needed.
Specific issues:
1. In figure 1 B, the levels of LVEF, LVFS, LVID; d, LVID; with G-Rb3, G-Rb3-FGL treatment were recorded by echocardiography. Not all the data showed that the combination group was better, please discuss why the FGL group is more effective than the combination treatment group.
2. In figure 2 D, western blot showed that G-Rb3 and FGL could promoted the expression, but the FGL group was better than the combination group. Does it mean that FGL is better in this aspect?
3. The effect of FGL on P-gp protein should be supplemented.
4. It is necessary to supplement the effect of single G-Rb3 treatment on Rh123.
5.Detection of P-gp protein alone does not demonstrate increased bioavailability.

Author Response
This work examines the efficacy of G-Rb3-FGL co-treatment in improving cardiac function 72 after MI were investigated in vivo. The pharmacological effects of G-Rb3-FGL on RXRα- 73 Nrf2 pathway were explored in vivo and in vitro. Through left anterior descending branch (LAD) coronary artery ligation or oxygen- 22 glucose deprivation-reperfusion (OGD/R) were used to establish MI models in vivo and in vitro. To verify the pharmacological effects and mechanisms of G-Rb3-FGL were explored by in vitro 24 studies.
This research is both practical and novel. But, some of the data in the paper are confusing, so several modifications and explanations are needed.
Point 1: 1.In figure 1 B, the levels of LVEF, LVFS, LVID; d, LVID; with G-Rb3, G-Rb3-FGL treatment were recorded by echocardiography. Not all the data showed that the combination group was better, please discuss why the FGL group is more effective than the combination treatment group.
Response 1: Thank you for your valuable comment. The EF value and FS value are important indexes to evaluate cardiac function injury. As shown in figure 1B, the EF value was improved after both G-Rb3 (p < 0.001) and G-Rb3-FGL (p < 0.001). Notably, there was statistical significance between G-Rb3 and G-Rb3-FGL (p < 0.01). Similarly, the FS value was also increased after G-Rb3 (p < 0.01) and G-Rb3-FGL (p < 0.001) treatment, and there was statistical significance between G-Rb3 and G-Rb3-FGL treatment groups (p < 0.05). Most importantly, the dose of G-Rb3 in this study was 0.63mg/kg/d, while the dose of G-Rb3 in the G-Rb3-FGL group was 0.32mg/kg/d. We apologize for any confusion our oversight may have caused you.
Point 2: In figure 2D, the western blot showed that G-Rb3 and FGL could promote the expression, but the FGL group was better than the combination group. Does it mean that FGL is better in this aspect?
Response 2: Thank you for your comment. As shown in figure 2D, the expression of RXRα increased after treatment with G-Rb3 (p < 0.05) but increased more significantly in the G-Rb3-FGL (p < 0.01) treatment group. The expression trend of the other proteins is the same. Notably, the treatment of G-Rb3 alone neither increase the expression of Nrf2 (p >0.05) nor decrease the expression of p65-NF-κB (p >0.05), but G-Rb3-FGL treatment could produce a statistically significant difference in these proteins (p < 0.01 and p < 0.05). These results indicated that the injury caused by MI could be better improved after G-Rb3-FGL treatment. Thank you again for your comment. We hope our explanation could solve your question.
Point 3: The effect of FGL on P-gp protein should be supplemented.
Point 4: It is necessary to supplement the effect of single G-Rb3 treatment on Rh123.
Response 3: Thank you for your comments. In order to better clarify our research methods, we combined the explanations for your comment 3 and comment 4. Actually, there was a gradual deepening process in our study. Our previous work [1-2] found that the effective concentration of G-Rb3 in the cell experiment was 40μM, but in this study, the dose of G-Rb3 was reduced due to the addition of FGL. In the present study, we focused on the effect of FGL-G-Rb3 on P-gp to explore the potential synergistic effects of FGL with G-Rb3 on P-gp. Meanwhile, the staining of Rh123 by FGL was carried out in figure 7C to indirectly illustrate the relationship between FGL and P-gp. Then through figure 7B, we further verified the relevant conclusions that FGL may be the main reason for the decreased expression of G-Rb3 on P-gp. Let’s put it another way, the fluorescence staining of Rh123 performed in figure 7C was to better clarify the views of figure 7B.
We planned to gradually carry out further research on this section. Unfortunately, due to the impact of COVID-19, the experimental platform has been closed for some time now. As a result, the above in-depth research may not be completed in the short term. We feel deeply sorry that we are unable to turn in the relevant supplementary data to you within the specified time.
Hope our explanation works for you. Relevant references are as follows:
References
[1] Chen X, Wang Q, Shao M, Ma L, Guo D, Wu Y, Gao P, Wang X, Li W, Li C, Wang Y. Ginsenoside Rb3 regulates energy metabolism and apoptosis in cardiomyocytes via activating PPARα pathway. Biomed Pharmacother. 2019 Dec;120:109487. doi: 10.1016/j.biopha.2019.109487. Epub 2019 Sep 29. PMID: 31577975.
[2] Shao M, Guo D, Lu W, Chen X, Ma L, Wu Y, Zhang X, Wang Q, Wang X, Li W, Wang Q, Wang W, Li C, Wang Y. Identification of the active compounds and drug targets of Chinese medicine in heart failure based on the PPARs-RXRα pathway. J Ethnopharmacol. 2020 Jul 15;257:112859. doi: 10.1016/j.jep.2020.112859. Epub 2020 Apr 12. PMID: 32294506.
Point 5: Detection of P-gp protein alone does not demonstrate increased bioavailability.
Response 4: Thank you for pointing out this issue. It is known that the detection of oral bioavailability is an important step for the new drug to successfully enter into the pre-clinical development stage, and we agreed that P-gp cannot be used as the sole criterion for the evaluation of oral bioavailability [1-2]. In fact, our detection of P-gp in this study is more like inspired research, and provides our team with guidance to carry out related experiments in the future. We apologized for the inappropriate description of experimental results in this section, and we have modified it both in the “Results” section and in the “Discussion” section of the paper.
In the end, we’re happy to edit the text further based on your helpful comments and we sincerely hope that the description in our article is now easier to follow with this new version. Relevant references are as follows:
References
[1] Elmeliegy M, Vourvahis M, Guo C, Wang DD. Effect of P-glycoprotein (P-gp) Inducers on Exposure of P-gp Substrates: Review of Clinical Drug-Drug Interaction Studies. Clin Pharmacokinet. 2020 Jun;59(6):699-714. doi: 10.1007/s40262-020-00867-1. PMID: 32052379; PMCID: PMC7292822.
[2] Zhang H, Xu H, Ashby CR Jr, Assaraf YG, Chen ZS, Liu HM. Chemical molecular-based approach to overcome multidrug resistance in cancer by targeting P-glycoprotein (P-gp). Med Res Rev. 2021 Jan;41(1):525-555. doi: 10.1002/med.21739. Epub 2020 Oct 12. PMID: 33047304.

Round 2
Reviewer 3 Report
This study may be considered for acceptance